# Investigating Smart City Development Based on Green Buildings, Electrical Vehicles and Feasible Indicators

**Armin Razmjoo [1,\*], Meysam Majidi Nezhad [2], Lisa Gakenia Kaigutha [3], Mousa Marzband [3,4], Seyedali Mirjalili [5,6], Mehdi Pazhoohesh [7], Saim Memon [8], Mehdi A. Ehyaei [9] and Giuseppe Piras [2]**

1   Escola Técnica Superior dÉnginyeria Industrial de Barcelona (ETSEIB), Universitat Politécnica de Catalunya (UPC), Av. Diagonal, 647, 08028 Barcelona, Spain
2   Department of Astronautics, Electrical and Energy Engineering (DIAEE), Sapienza University of Rome, 00185 Roma, Italy; Meysam.majidinezhad@uniroma1.it (M.M.N.); giuseppe.piras@uniroma1.it (G.P.)
3   Department of Math's, Physics and Electrical Engineering, Faculty of Engineering and Environment, North Umbria University Newcastle, Newcastle upon Tyne NE1 8ST, UK; gakenia.kaigutha@northumbria.ac.uk (L.G.K.); mousa.marzband@northumbria.ac.uk (M.M.)
4   Center of Research Excellence in Renewable Energy and Power Systems, King Abdul-Aziz University, Jeddah 21589, Saudi Arabia
5   Centre for Artificial Intelligence Research and Optimisation (AIRO), Torrens University Australia, Adelaide, SA 5000, Australia; ali.mirjalili@gmail.com
6   YFL (Yonsei Frontier Lab), Yonsei University, Seoul 03722, Korea
7   School of Engineering and Sustainable Development, De Montfort University, The Gateway, Leicester LE1 9BH, UK; mehdi.pazhoohesh@dmu.ac.uk
8   Solar Thermal Vacuum Engineering Research Group, London Centre for Energy Engineering, School of Engineering, London South Bank University, 103 Borough Road, London SE1 0AA, UK; s.memon@lsbu.ac.uk
9   Department of Mechanical Engineering, Pardis Branch, Islamic Azad University, Pardis New City 1468995513, Iran; aliehyaei@yahoo.com
\*   Correspondence: arminupc1983@gmail.com

**Abstract:** With a goal of achieving net-zero emissions by developing Smart Cities (SCs) and industrial decarbonization, there is a growing desire to decarbonize the renewable energy sector by accelerating green buildings (GBs) construction, electric vehicles (EVs), and ensuring long-term stability, with the expectation that emissions will need to be reduced by at least two thirds by 2035 and by at least 90% by 2050. Implementing GBs in urban areas and encouraging the use of EVs are cornerstones of transition towards SCs, and practical actions that governments can consider to help with improving the environment and develop SCs. This paper investigates different aspects of smart cities development and introduces new feasible indicators related to GBs and EVs in designing SCs, presenting existing barriers to smart cities development, and solutions to overcome them. The results demonstrate that feasible and achievable policies such as the development of the zero-energy, attention to design parameters, implementation of effective indicators for GBs and EVs, implementing strategies to reduce the cost of production of EVs whilst maintaining good quality standards, load management, and integrating EVs successfully into the electricity system, are important in smart cities development. Therefore, strategies to governments should consider the full dynamics and potential of socio-economic and climate change by implementing new energy policies on increasing investment in EVs, and GBs development by considering energy, energy, techno-economic, and environmental benefits.

**Keywords:** smart cities; policy; green buildings; electric vehicles; indicators

## 1. Introduction

With the growth in population and frequent energy crisis, moving toward sustainable cities is a political and important plan for many countries [1]. In this regard, the key

role of energy is remarkable, because it plays a significant role in the economic and social development of countries [2]. Green energy technologies are a rapidly growing reliable source of power, which can mitigate the energy crisis while conserving the environment [3]. Essential renewable energy applications in smart city development are for green buildings (GBs) and electric vehicles (EVs). Limited resources of fossil fuels and an increase in demand for electricity have renewed interests in integrating GBs [4] and moving towards EVs in urban areas [5].

Several studies exploring different aspects of smart cities, including smart cities, utilization of EVs, and the importance of GBs for smart cities, have been investigated. For instance, Danuta Szpilko et al. (2019), argued the different advantages of smart cities: urban flexibility, appropriate consumption patterns, environmental issues, and clean energy utilization. They proved that smart city development has a lot of positive impacts on future cities and countries [6]. Roberto Ruggieri et al., based on the 11th goal of the SGDs by 2030, investigated the management of existing cities and the planning of future ones, based on Sustainable Cities and communities regarding energy consumption and decarbonization of transport. In these regards, they investigated electric mobility performance in six Smart Cities (Oslo, London, Milan, Hamburg, Bologna and Florence). The analysis showed that electric vehicles' use was positively affected by reducing PM2.5, PM10, and NO2. In this respect, the cities demonstrated the most remarkable reduction in pollutant (above 20%) were Hamburg (−28% PM2.5 and −2%6 NO2), Milan (−25% PM2.5 and −52% NO2), and London (−26% NO2) [7]. To increase the power grid's peak load by uncoordinated charging of EVs and an increasing extra expense of electricity, Zhang et al., proposed a new strategy based on three aspects: coordination, practicality, and autonomy.

In addition, this strategy includes a day-ahead pricing scheme, which can overcome the minimum optimization expense [8]. Due to the increasing charging demand of EVs, Kamankesh et al. suggested optimal scheduling of GBs. The optimal energy management for GBs, including renewable energy sources (RESs), plug-in hybrid electric vehicles (PHEVs), and storage devices, was discussed. They introduced different charging methods through uncontrolled, controlled and smart charging strategies [9]. Honarmand et al., proposed an energy resource management model for GBs to overcome issues related to integrating EVs, RES and, power networks. This method was able to investigate practical limitations, anticipating errors of renewable energy in the line of EVs owner satisfaction [10]. Peter K. Joseph et al., using renewable energy, examined vehicle-to-grid, and wireless charging integration of electric vehicles using renewable energy for sustainable transportation.

This research, which was conducted in the pursuit of energy conservation, particularly in developing countries, demonstrated that combining electric vehicles with renewable energy sources can result in parking spaces being transformed into unlimited sources of clean energy and preventing energy loss [11]. Anber Rana et al., investigated financial Incentives (FIs) for green buildings for Canada. They showed that FIs for buildings in Canada can be distributed into four categories: grants, tax, loans, and rebates, and among these, rebates are the most common in all provinces. They mentioned that these incentives belong to three end-users (aboriginal people, landlords and tenants, and low-income) and for three types of buildings (energy rated, heritage, and non-profit). With these incentives, four provinces (Ontario, Alberta, British Columbia, and Quebec) are leading green building efforts [12].

Zhen Liu et al., examined the potential impact of the integration of building information management and blockchain on making buildings more sustainable in a smart city environment. They showed that a complete life cycle study could help constructors, designers, supervisors, and decision-makers make informed and accurate decisions for green buildings in smart city development [13]. The importance of the Internet of Things (IOT) as an intelligent technology, examined by Waleed Ejazthe et al., enabled electric vehicles in smart cities. Actually, and due to this reality, the large-scale implementation of electric vehicles can make extra burdens on electric grids; thereby, they suggested smart

scheduling using IOT technology to optimize smart cities' charging process. Hongyu Chen et al. [14], evaluated the environmental aspects of green buildings in smart cities based on the Internet of Things system. They showed that using IoT can reduce the negative environmental aspects in green buildings [15].

### 1.1. Motivation and Objective of the Study

Energy consumption is likely to rise dramatically by 2050 as a result of predicted high population expansion, particularly in metropolitan areas. The expected exponential expansion emphasizes the significance of researching and deploying smart technologies in order to fulfill demand. Smart cities are critical for maintaining supply and demand equilibrium. However, it will not be possible until governments throughout the world adopt net-zero energy policy. Therefore, this study provides a comprehensive review of smart cities, including an introduction and literature review, investigating significant barriers and introduces feasible indicators for further analysis in this area. A fundamental component of a smart city is net-zero energy GB. In addition, electric vehicles provide essential benefits to smart cities, including $CO_2$ emission reduction.

Therefore, evaluating the GBs, EVs, and their correlations in smart cities is the main motivation of this study. Currently, EVs are presented as eco-friendly for the environment and reliable power backup systems in the transport sector. They can mitigate high percentages of $CO_2$ emissions, which is better for the environment for they run on electrical energy instead of combustible fuels. Therefore, this type of technology creates an appropriate opportunity to utilize them as storage systems. By increasing the use of renewable energy sources, the capacity of the energy storage system is increased to control power generation and fluctuation. On the other hand, creating, and developing GBs, is a positive step for reduction of $CO_2$ emissions, and providing energy for the citizens of smart cities.

### 1.2. Hierarchical Structure of Smart Cities

Figure 1 shows the hierarchical structure of application areas related to smart cities.

Additionally, Table 1 shows the previous investigations related to EVs and GBs by different researchers. As can see the key role of renewable energy for smart city development is significant in these studies.

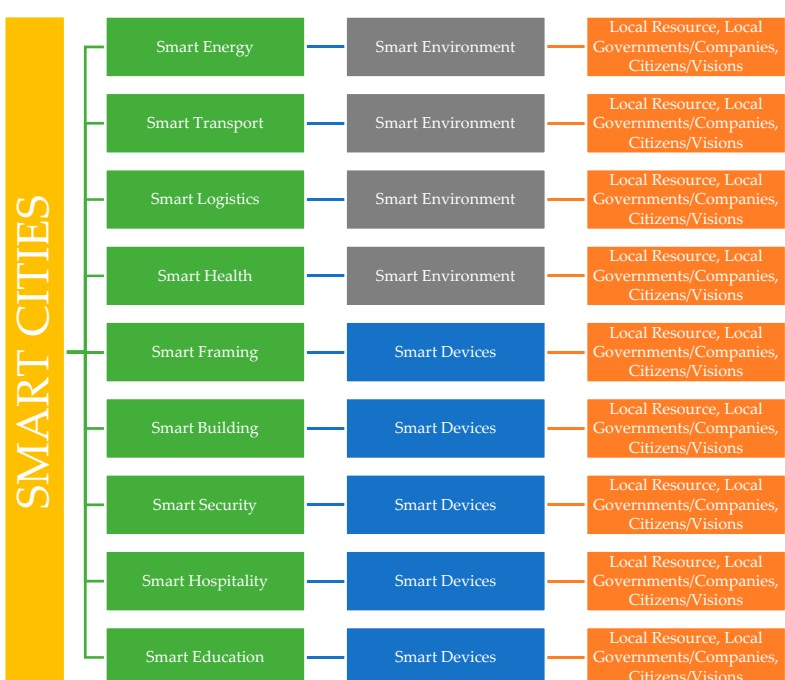

**Figure 1.** Shows the hierarchical structure of application areas related to smart cities [16].

**Table 1.** The previous investigations related to EVs and GBs by different researchers.

| Summery Description of the Work | Reference |
|---|---|
| Investigation on the utilization of proper planning for city development using optimal energy technology networks and policy. | [17] |
| Investigating the challenges and solutions of energy Management using the Iof T in Smart Cities. | [18] |
| Presentation of practical standards (potential of ZEBs in SET-Plan smart cities) in the field of energy and for GB. | [19] |
| Examination of electric vehicle integration in green smart cities with IOT. | [20] |
| Consideration city-integrated using renewable energy for urban sustainability. | [21] |
| Investigation sustainability of city that can be as energy efficient low carbon zones. | [22] |
| Investigation carrier networks in urban areas for distributed renewable energy generation to improve energy sustainability. | [23] |
| Consideration of user-centric smart GBs to achieve energy sustainability for smart city. | [24] |

## 2. Importance of GBs and EVs

### 2.1. Green Building Development

According to research work published in 2021 by Yu-HaoLin et al., the green buildings have a considerable effect on annual building energy efficiency, and reduction of $CO_2$ emissions, and these positive effects will be significantly increased shortly because of the growing population and rural to urban migration [1]. Therefore, one of the best ways to manage energy consumption and $CO_2$ emission is by implementing GBs. GBs using photovoltaic panels installation to provide the energy required and utilize natural resources in the urban areas, which can reduce the negative impacts on climate change. Therefore, GBs (using photovoltaic system installation) can supplement the amount of energy required by consumers and decrease the dependency on fossil fuels, which reduces global warming [25].

### 2.2. The Importance of Electric Vehicles (EVs)

EVs have a crucial role in developing smart cities [26]and minimizing $CO_2$ emissions [26]. In this case, EVs that use renewable energy could be an effective solution to achieve these goals. Renewable energy sources have a remarkable role in yielding better power quality in distribution systems [27]. On the other hand, integrating EVs with renewable energy sources reduces a notable amount of carbon emissions that are important for the environment's future sustainability [28].

Consequently, the application of hybrid electric cars cuts fuel consumption, conserves energy and runs cleaner, which reduces adverse environmental effects, especially in the transport sector [29]. EVs linked to charging stations in GBs, is a significant energy resource in the energy management system. Although, for an optimal energy management system, an independent controller design is necessary to plan the charging and discharging steps of the battery [30]. Fortunately, the charging infrastructure has been developed in several countries such as North America, Japan, Europe, and China. Considerable efforts for the development of EV charging infrastructure have been carried out to implement effective indicators in the community, such as energy demand management from EVs, managing energy intensity, and considering environmental impacts like charger's intensity distribution and carbon intensity [31]. Undoubtedly, as more and more electric vehicle models are produced and are available on the market, the number of EVs sold and produced will also grow in Europe.

According to a forecast regarding electric vehicles in Europe, the number of EVs produced as Figure 2, from around three quarters of a million in the 2019 year to more than 4 million in the 2025 year [32].

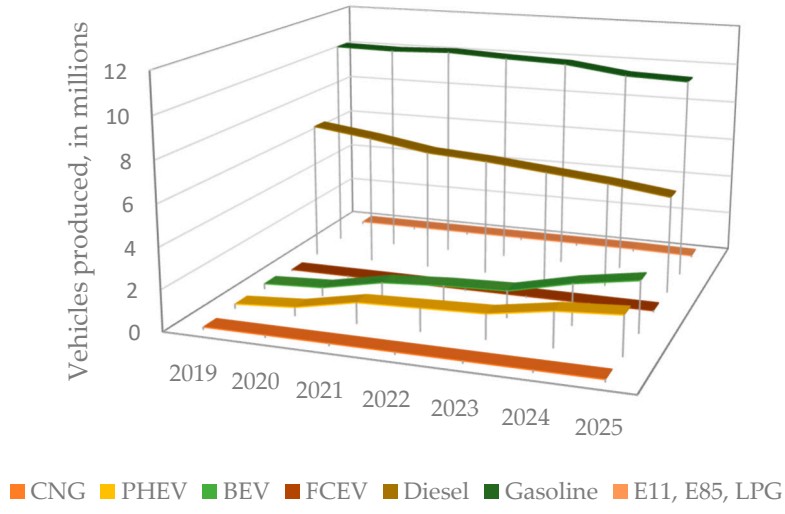

**Figure 2.** EU production of vehicles per type in 2025, in million units [32].

### 2.3. Importance of Quality with Expansive Use of EVs

It is evident that in a few years, EVs are going to replace internal combustion engine vehicles and are more favorable than in the past [33]. To achieve this target, two essential factors have been considered: quality and cost [34]. Globally, manufacturers and researchers active in the EVs field [35] are working on new technology regarding the private transport sector and building sector in smart grid energy districts. [36]. Additionally, assessment impacts of the potential of plug-in vehicles have been increased [37]. Different companies are developing strategies to reduce the cost of production of EVs while maintaining good quality standards. Renowned vehicle manufacturers are increasing their investments to offer a broader range of EV models in various sizes. However, the choice of EV designs is still limited compared to conventional vehicles [32].

### 3. Methodology

In this study, smart cities examined comprehensively with emphasis on policy, GBs, and EVs impacts. To obtain the information required for this work, in the first stage, we used "smart city", policy and strategy for smart cities, renewable energy, barriers, and solutions for smart cities development," as titles, abstracts, and keywords in the query entry and started the search process by using established scientific databases, such as Scopus, Google Scholar, Web of Science and journal sites (Elsevier, Springer, Tylor & Francis, MDPI, Willey, etc.). In this stage, every author collected related research for the corresponding author and sent it. Then, and based on the eligibility criteria and their accessibility, we have, over three years, identified and conducted an exhaustive review of more than 300 relevant publications and scientific reports related to smart cities such as EU Smart Cities Marketplace, governmental reports and European energy reports. After the collection information stage, we evaluated the journals based on titles, abstracts, and introductions and selected the appropriate articles to collect 98 articles. Next, we did two stages in parallel:

(a) Reviewing reports and review articles to have a global understanding of smart cities development issues and find the proper solutions to overcome these problems. These studies helped us improve our background and knowledge for writing the manuscript.

(b)   Reviewing technical articles. These articles were fruitful in identifying appropriate policies to address barriers in the development of smart cities and defining the correct pathway for the study.

## 4. Results and Discussion

Confronting the essential issues such as climate change and energy crisis for the people in the future using creating smart cities can be achieved. In this regard, the key role of GBs and EVs and correlations between them in the line of smart cities development is crucial. Thereby, this research investigates smart cities development based on two of these crucial sections. Indeed, this paper has several novelties, including investigating different aspects of smart cities development, presenting the main barriers with relevant solutions to overcome them for smart cities development, and introduces new feasible indicators related to GBs and EVs in designing SCs. This section discusses achievable policies, recognizing barriers with the solutions in the line of smart cities development, and efficient indicators for GBs and EVs.

### 4.1. Crucial Role of Policy for Creation and Development Smart Cities

Undoubtedly, one of the most critical sectors for progressing and developing smart cities is policy [38]. Additionally, the policy has a decisive role in advancing the goals of governments [39]. The use of the appropriate policies and strategies by policymakers leads to effective and good results for the development of smart cities [40]. However, implementing appropriate policies in societies is not easy and requires public support. To do this, the diversity of stakeholders' rationales to implementing participatory processes should be investigated using proper methods and correct instruments for participation [41].

As active involvement is a key aspect in achieving success, policymakers should include stakeholder expectations into decision-making in order to successfully implement policies and avoid public resistance [42].

In this sense, elected leaders must work to ensure that inhabitants have the finest possible living conditions in future cities. Policies such as the use of smart technology (IoT) to maximize efficiency and reduce costs, developing GBs, developing smart meters systems in buildings, the integration of connected local energy storage systems to improve quality of life and foster economic development, the development of electric vehicles (EV) systems in cities and encouraging people to use these vehicles more, strict monitoring of electric energy companies, and zero energy buildings are just a few examples.

### 4.2. Existing Barriers against Smart Cities Development

Undoubtedly, recognizing main issues as mentioned above can help national governments and local governments in developing SC. Thereby, in this section, the main barriers against the progress of smart cities are presented and then discussed. We believe the main issues related to smart cities development are in this table and can be a good guide for researchers in the future. Therefore, with consideration of these barriers, policymakers will be able to find the appropriate solutions for them that lead to accelerating smart cities development. These barriers as a set cover small and big issues that policymakers are looking to overcome in SC development. These barriers are divided into technical, environmental, economic, social, and governmental categories in Table 2 and are including the barriers of governance, social, technology, environment, and economy. Each of these, as following, categories also includes other vital barriers, the impact of which can be removed or decreased as necessary to ease the creation and development of smart cities.

**Table 2.** Categories of the key barriers against the development of smart cities.

| | |
|---|---|
| 1. Weak cooperation between urban planners and policymakers (G) | 12. Defective system (T) |
| 2. Poor information technology management (G) | 13. Weak information technology infrastructure (networks) (T) |
| 3. Improper regulatory norms and policy (G) | 14. Less use of clean energy (EN) |
| 4. Weak public-private participation (G) | 15. Weak interaction between citizens and local governments (S) |
| 5. Lack of proper strategies for development (G) | 16. Insecurity of energy sustainability (EN) |
| 6. Irresponsible citizens (S) | 17. Not paying attention to environment (EN) |
| 7. Lack of attention to public welfare include entertainment places and parks (S) | 18. Weak and inappropriate IT infrastructures (EC) |
| 8. Weak communication of citizens and low knowledge (S) | 19. Weak training of the public (EC) |
| 9. Inadequate geographic and environmental assessment before the build of smart cities (S) | 20. Higher maintenance and operational cost (EC) |
| 10. Inequality and discrimination (S) | 21. Inappropriate plans in order to attract foreign investments (EC) |
| 11. Inappropriate access to new technology (T) | 22. Not paying attention to stakeholders participation (S) |

However, the main question is why these barriers are essential, and how can overcome them. The policymakers and energy experts should have appropriate policies based on recognizing the main barriers against smart cities development and reducing and removing these barriers. The policies that lead to improving transportation networks, more use of EVs, reduction of $CO_2$ emission, developing GBs, energy-saving, more use of renewable energy, energy managing by the smart meter especially for buildings, affordable energy, improvement of IT networks, and increasing of green spaces. Therefore, the barriers presented cover the main issues in this way and are analytical. Table 3 shows the relevant references for these indicators.

**Table 3.** Relevant references for the above barriers.

| Category | Related References |
|---|---|
| Governance | [43–66] |
| Social | [42–46,59,67,68] |
| Technology | [43–45,48,49,69–75] |
| Environment | [43,46,47,76–79] |
| Economy | [46,69,70,79–81] |

*4.3. Required Indicators*

The indicators for smart cities focus on monitoring the evolution of a city towards an even more innovative city. Therefore, the appropriate indicators for smart cities regarding the importance of the GBs and EVs have been considered. Policymakers and energy experts will be able to use energy sustainability indicators to measure its viability, solve existing problems and improve the weak points connected with the smart cities development. Therefore, the indicator as a time component for the development of smart cities over the years is a significant feature. Furthermore, it means that recognizing proper and efficient indicators can show us to what extent overall policy goals have been reached or are within reach.

The most important indicators regarding GB and EVs that affect energy sustainability in smart cities are presented in Table 4, which shows the indicators and sub-indicators for GBs and EVs [82–94]. The implementation of these indicators presented leads to the improvement of the residential life quality of the smart cities in the future. These indicators

cover all essential requirements for GBs and EVs development in smart cities. For example, in this Table, energy consumption, energy efficiency, energy saving, environmental adaptability, mentioned as indicators for both GBs, and EVs are essential, and implementation of these indicators on the GBs, and EVs leads to facilities and satisfaction of the citizens in the smart cities.

**Table 4.** The vital indicators for GBs and EVs.

| Indicators | GBs | EVs |
|:---:|:---:|:---:|
| Investment | √ | √ |
| Access electricity | √ | √ |
| Energy efficiency | √ | √ |
| Affordable price | √ | √ |
| Electricity consumption | √ | √ |
| $CO_2$ emission | √ | √ |
| Thermal comfort | √ | X |
| Speed | X | √ |
| Energy affordability | √ | √ |
| Smart meters | √ | X |
| Proper infrastructure | √ | √ |
| Renewable energy production | √ | X |
| Indoor air quality | √ | X |
| Energy conservation | √ | √ |
| Lighting efficiency | √ | √ |
| Renewable energy use | √ | √ |
| Environmental Adaptability | √ | √ |
| Architectural flexibility | √ | √ |
| Access to public services | √ | √ |
| Safety | √ | √ |
| Environmentally friendly design | √ | √ |
| Water management | √ | X |
| Waste management | √ | X |

*4.4. Finding Gaps and Solutions to Overcome and Confront with Barriers of Smart Cities Development and Present the Solutions*

Different aspects of smart cities are investigated in this study concerning the GBs, EVs, which also cover environmental challenges, smart city concepts, and smart city development hurdles. Undoubtedly, smart cities can be a conducive solution to overcome the most existing problems involving present humans. However, creating and developing smart cities will have problems for governments and countries. Now the question is what we should do. To answer this question, it would be great to mention correct policies and strategies especially in the line of GBs, and EVs development. As if these sections investigate properly and relevant issues of these become remove, many problems of citizens will be eliminated, thereby, all governments and countries need to think for finding proper solutions and ways.

Appropriate policy and strategy can be investigated before facing these problems by policymakers who influence governments' bodies. It means they can investigate and predict existing problems, and then present the best and logical solutions to overcome these problems.

Table 5 shows the barriers to smart cities based on six crucial sections of the smart city. The results of this Table that obtained from the references of Table 3, can be a good plan for policymakers that recognize them and implement them to overcome the problems

related to smart cities development in the future. Indeed, this table shows how a city can convert to a smart city by using appropriate policies and strategies and showed the main barriers and solutions for each of them.

**Table 5.** The barriers and solutions of smart cities (please referee to nomenclature to more understand the words shorted in the table).

| Characters | Barriers | Solution |
|---|---|---|
| Smart people | Avoidance of society; Old technology, | New technology, More communication, |
| | Lack of knowledge, | Caring community, Racal harmony, |
| | Irresponsible community | Talented and Skilled people |
| Smart governance | Low budget, Old technology, Poor | Appropriate policy and Strategy, New |
| | private-public participation, Incorrect | technology, Correct legislation, Public |
| | legislation (Policy and Strategy), | participation, Establishing Equality and |
| | Discrimination and inequality | Justice |
| Smart economy | Inefficient financial support, Insufficient | Attract investment, Entrepreneurship, |
| | Investment, Unemployment | Innovative economic, Equitable wealth |
| | | Distribution |
| Smart mobility | Weak ICT infrastructure, Weak public | More use of IT, Development internet |
| | transport, Lack of sufficient green spaces, | infrastructure, Green spaces development, |
| | Lack of sufficient transport, Lack of | Efficient road and accessibility, Public |
| | proper resiliency, Lack or improper of | participation, Improve Energy intensity |
| | traffic management system | |
| Smart environment | Use of fossil resources, Lack of sufficient | More use of clean energy, Green spaces |
| | sanitation and water, Lack or a few green | development, Sufficient sanitation and water |
| | Spaces | , utilization of electrical vehicles |
| Smart living | Lack of sources and internet access, | Sufficient information technology, New |
| | Insufficient information technology, Old | technology, Electrical vehicles, Utilization |
| | Technology | of robots, Safety and security information |
| | | Development |

Therefore, this Table, with a deeper view of the existing barriers concerning smart cities development, present efficient solutions in this regard. Although we believe implementing these solutions is not easy but given them for policymakers activating in smart cities development is a benefit. For instance, skilled and talented people in proper places can prevent future issues, leading to a progressive acceleration in different smart cities. In addition, more use of new technology like IT systems in smart cities development is an essential factor. Therefore, special attention to these barriers and solutions is significant for policymakers.

As Table 5 mentioned, the barriers and solutions of smart cities, therefore can be added for a better understanding of presenting this table that a city to become a smart city needs to essential changes. The changes such as in technology, type of communication, use of skilled and talented people in proper places, encourage public in participation, utilization of electrical vehicles, development of internet infrastructure, development of green spaces to improve the environment in cities, improve energy affordability especially through the use of clean energy, development GBs, providing sufficient water and sanitation, and improvement of the security information networks. Therefore, the aim of Table 5 is to recognize the main barriers and solutions for them in the line of smart cities development.

*4.5. Correlations between GB and EV in the Line of Smart Cities Development*

Regarding rapid urbanization and the development of the cities the issues such as energy crisis, climate change, and resource depletion are concerning for the people. Therefore, special attention to these issues and having appropriate planning to overcome these for all governments are necessary, otherwise, shortly the people become involved more with these crucial issues [95]. The two essential elements in creating sustainable cities are sustainable buildings [96] and electrical transportation systems [97].

The two main issues against creating, and developing sustainable cities, are the inefficiency of old buildings [98], and old transportation that uses fossil fuels as the main energy source [99]. In this case, the challenges are related to our total energy consumption and $CO_2$ footprint [100]. Thereby, more investment in the GBs and EVs for all policymakers and energy experts is significant and has an impressive impact on reducing the mentioned issues. In addition, if we will not have sustainable cities in the future, we need to develop GBs and EVs. As creating GBs, and EVs will reduce main social challenges such as loss of energy, $CO_2$ emissions, and in the end, livable and sustainable city for our self, and the next generation. Thus, there are close correlations between GBs and EVs in the line of smart cities development [101].

To complete this section, it can be added that the development of each of these sections has good results in smart cities. For example, in recent years the councils of the municipal cities have tried for the development of the green building in the line of smart cities as creating an energy-efficient world. It means that green buildings lead to environmentally friendly cities especially with consideration ecological materials that are used in building them. For this, the sustainable building councils, using the skilled teams of scientists, architects, and contractors a develop conceptual targets in the strict sense of sustainable building that new buildings become built based on the ecological, energy-saving, social-cultural, and economic concepts with high functional quality. In addition, these houses can have a low amount of energy consumption while providing satisfaction to residents, advanced life cycle assessment via the use of natural materials and concepts, capacity to deploy renewable energy sources (solar panels, wind turbines) in their roofs, and higher profitability via reduced energy consumption. On the other hand, these policymakers, and energy experts for implementing charge places for electric vehicles, and development of them have many tried that have succeeded to a great extent [102–104].

Thereby, for explanation more accurate of the correlations between GBs and EVs in the line of smart cities, and development, can be said that since the aim of the smart cities'

development is make more welfare of citizens while conserving the environment, energy affordable, clean, and abundant for them.

## 5. Conclusions and Recommendations

The primary purpose of this research was to study existing barriers of smart cities development with identify appropriate solutions for each of them and present new feasible indicators related to GBs and EVs. After a comprehensive review of existing barriers and solutions for smart city development, new indicators have been presented that show the positive impact on designing and developing smart cities. First and foremost, this study addresses SC facility development in the context of the increasing prevalence of EV and GB in smart cities. In this regard, governments must have a coherent strategy for growth in EV and GB to develop smart cities, have adoption environmental methods to reduce CO2 emission, and use key demographic indicators that cause the implementation process.

The findings revealed that, in the 2020s, improvements in energy and resource efficiency will play a particularly important role in decreasing industrial emissions, paving the path for broad emissions reductions while infrastructure for profound decarbonization alternatives that can be built up for which new ideas and advancement in GBs and EVs and introduction to economic and financial benefits to consumers would be a key motivation in accelerating in achieving SCs. Additionally, the findings revealed that implementing strategies and embracing change in government policies would help to accelerate the net-zero energy vision, pay attention to design parameters, improve transportation policies while embracing change, implement efficient indicators for GBs and EVs, and implement strategies to reduce the cost of EV production while maintaining high quality. As a result, governments should have appropriate policies in place, such as boosting investment in EVs and developing GBs in terms of environmental approaches.

Without a doubt, this work has limitations like many similar works published that other researchers can consider in the future. The limitations are related to the causes of the lack of enough governments' investments in GBs, and EVs development in cities, causes of the lack of proper cooperation between national governments with local governments in implementing the targets, etc. Therefore, recognizing the relevant political issues, and finding the efficient solutions for them, can be investigated by those interested in working in this regard.

**Author Contributions:** A.R., M.M.N., Investigation, Visualization, Data curation, writing and methodology, L.G.K., M.M.; Investigation, S.M. (Seyedali Mirjalili), M.P.; Resources, Formal analysis, S.M. (Saim Memon), M.A.E., and G.P.; Writing—review and editing. All authors have read and agreed to the published version of the manuscript.

**Funding:** Not Applicable.

**Institutional Review Board Statement:** Not Applicable.

**Informed Consent Statement:** Not Applicable.

**Data Availability Statement:** Not Applicable.

**Acknowledgments:** The authors would like to express their appreciation for the international research collaboration in Renewable Energy.

**Conflicts of Interest:** The authors declare no conflict of interest.

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
