# Peer review of "Investigating Smart City Development Based on Green Buildings, Electrical Vehicles and Feasible Indicators"

_sustainability, doi:10.3390/su13147808_

Round 1
Reviewer 1 Report
The paper has a research on Investigating smart city development based on green Buildings, Electrical Vehicles and Feasible Indicators.
The presented research is an interesting case. The authors did a good job in the revision.
Nonetheless, the work has several flaws, as summarized below.
1. From the current version of the manuscript, it is hard to tell whether there is a novel scientific impact.
2. There are lots of repetitions in the paper.
3. In the conclusion authors may briefly discuss the potential limitations of the paper and what are the future research directions of this study. In other words, how other researchers can work on this study to continue this line of research?
Author Response
Thank you, dear Reviewer

Reviewer 2 Report
Thank you for this resubmission. I can see now that this paper is in fact a literature review, which was not made clear in the original submission.
The authors made significant changes to the paper. However, there are still some of its aspects that need revisions.
- L 117 : The authors claim the main motivation of the study is the correlations of green buildings, electric vehicles in smart cities. What does that mean in this context? In statistics, correlations mean that two variables are related. Following that logic, results should show the connections between green buildings and electric vehicles in smart cities, from technical, regulatory etc. perspectives. But when looking at the results section, there is no mention of the “correlations” between green buildings and electric vehicles. In particular, tables 2 and 5 are general to smart cities.
- The paper lacks a critical perspective that is usually common to literature reviews. For example, section 4.3 on the “required indicators” is minimal in terms of explanations. Indicators are presented in a table but with no argumentation as to why they are here, why they are important to smart cities’ viability etc. Are they important in every context, every city? How can cities measure them? For example, the link between “improve the weak points in connection with the central energy system” (line 278) and the indicator “waste management” is not clear.
- A consequence of the previous comment is that the results or findings are very generic and can’t be used as they are to help decision-makers. For example, a solution proposed to “lack of sufficient green spaces” is “green spaces development” or a solution to “discrimination and inequality” is “equality and justice”.
- I would suggest reducing the scope of the analysis and focusing on the “correlations” between GB and EV as initially intended, which is a worthy object of study because potentially useful for decision-makers (who may see these two solutions independently). Results and findings are presently too generic, focused on smart cities as a whole and with no developed argumentation.
Author Response
Thank you, dear reviewer

Reviewer 3 Report
Thank you for addressing the comments
Author Response
Thank you, dear Reviewer
